# Roles of Interferon Regulatory Factor 1 in Tumor Progression and Regression: Two Sides of a Coin

**DOI:** 10.3390/ijms25042153

**Published:** 2024-02-10

**Authors:** Alina M. Perevalova, Lyudmila F. Gulyaeva, Vladimir O. Pustylnyak

**Affiliations:** 1Zelman Institute for the Medicine and Psychology, Novosibirsk State University, Pirogova Street, 1, Novosibirsk 630090, Russia; a.perw@yandex.ru (A.M.P.);; 2Federal Research Center of Fundamental and Translational Medicine, Timakova Street, 2/12, Novosibirsk 630117, Russia

**Keywords:** IRF1, interferon, cancer, immunotherapy

## Abstract

IRF1 is a transcription factor well known for its role in IFN signaling. Although IRF1 was initially identified for its involvement in inflammatory processes, there is now evidence that it provides a function in carcinogenesis as well. IRF1 has been shown to affect several important antitumor mechanisms, such as induction of apoptosis, cell cycle arrest, remodeling of tumor immune microenvironment, suppression of telomerase activity, suppression of angiogenesis and others. Nevertheless, the opposite effects of IRF1 on tumor growth have also been demonstrated. In particular, the “immune checkpoint” molecule PD-L1, which is responsible for tumor immune evasion, has IRF1 as a major transcriptional regulator. These and several other properties of IRF1, including its proposed association with response and resistance to immunotherapy and several chemotherapeutic drugs, make it a promising object for further research. Numerous mechanisms of IRF1 regulation in cancer have been identified, including genetic, epigenetic, transcriptional, post-transcriptional, and post-translational mechanisms, although their significance for tumor progression remains to be explored. This review will focus on the established tumor-suppressive and tumor-promoting functions of IRF1, as well as the molecular mechanisms of IRF1 regulation identified in various cancers.

## 1. Introduction

The IRF1 protein (interferon response factor 1), a transcription factor belonging to the IRF family, is best known for its role in regulating immune responses [1]. IRF1, along with other IRFs, is thought to be involved in the cellular immune responses to pathogen-associated diseases, primarily those caused by RNA and DNA viruses [2]. Recent findings indicate that IRF1 may play a role beyond promoting cellular responses to pathogens. Much evidence has been discovered regarding IRF1’s function in regulating DNA damage-induced apoptosis and tumor cell growth arrest. Additionally, it has been demonstrated that IRF1 regulates immune cell development and differentiation, antitumor immune response, and several other carcinogenesis-related processes [3]. Despite ongoing research, the role of IRF1 in carcinogenesis remains a subject of conflicting information, and its impact on cancer development is still uncertain. In fact, several studies have demonstrated its antagonistic effects on tumor growth and antitumor immune response. Such inconsistencies suggest its involvement in maintaining the balance of pro- and antitumor mechanisms, both in tumor cells and in the microenvironment. Furthermore, IRF1 is believed to be a crucial mediator of pathways that involve specific “immune checkpoint” molecules, allowing us to envision its potential use in cancer immunotherapy. Nevertheless, all of the details of IRF1 molecular interactions and the conditions that govern them remain to be studied.

## 2. IRF Protein Family

The IRF family involves transcription factors which regulate signaling pathways induced by viruses, bacteria, and interferons of various classes. Nine of them are found in human cells: IRF1, IRF2, IRF3, IRF4 (PIP/LSIRF/ICSAT), IRF5, IRF6, IRF7, IRF8 (ICSBP), and IRF9 (ISGF3g/p48) [1,4].

Amino acid sequences in IRFs contain several homologous regions that are essential for protein function. All IRFs share homology in their N-terminal regions, which contain a DNA-binding domain (DBD) with five regularly spaced tryptophan residues [2,4]. The DBD is believed to have a role in recognizing ISREs (interferon-stimulated response elements), which are specific DNA sequences found in the promoters of various interferon-stimulated genes (ISGs), type I and type III IFN genes [2]. IRFs’ DNA-binding domain is responsible for recognizing DNA sequence elements similar to ISREs (^A^/_G_NGAAANNGAAACT) [5]. Structural analysis demonstrated two key regions within this sequence: the upstream AA region, recognized by His40 of IRFs’ L1 loop, and the downstream GAAA region, recognized by their α-helix 3. DNA regulatory elements containing GTGAAA hexanucleotide sequences were further named as IRF-binding elements (IRF-E). In addition to the ISREs, IRFs’ DBD can bind to other 5′-GAAA-3′-containing regions, such as PRD1 (positive regulatory domain I) in the IFN-*β* gene promoter [1,6].

Compared to the N-terminal regions of IRFs, their C-terminal regions are more diverse in structure and are thought to be responsible for controlling transcriptional activity. Each IRF has its own specific transcriptional activity, which is determined by its ability to interact with other IRFs, as well as with other different factors and co-factors. Such interactions are controlled by the IRF-association domain (IAD, or regulatory domain) located in the C-terminal regions of proteins. There are two types of such domains that have been identified: IAD1 and IAD2. IAD1 is present in all IRFs except IRF1 and IRF2, which, in turn, contain IAD2 in their structure [7]. Protein–protein interactions governed by these domains can modulate the activity of entire protein complexes located on target gene regulatory elements and determine DNA sequences near ISREs that can be bound by such complexes [1,8].

## 3. IRF1 Gene and Protein Structure and Splicing Variants

The human *IRF1* gene is located in the 5q31.1 locus, has a length of 5528 base pairs, and contains nine exons and eight introns [3,9]. Apart from its main transcript, several alternative splice variants are known for IRF1. A study of bone marrow and peripheral blood samples from patients with myelodysplastic syndrome and chronic myelocytic leukemia identified IRF1 variants lacking exon 2 or both exons 2 and 3 (IRF1Δ2 and IRF1Δ23). Such variants were unable to function as transcription factors due to the absence of a protein DNA-binding domain [10]. Other forms of IRF1, lacking exons 7, 8, 9, and their combinations, were identified in cervical cancer tissue samples (IRF1Δ7, IRF1Δ8, IRF1Δ9, and others). In this case, IRF1 isoforms were able to display activity in the cell, but their transcriptional activity turned out to be variable due to the absence of different parts of the regulatory domain. The absence of ubiquitin binding sites in these IRF1 variants resulted in a longer half-life and greater stability, making them more stable than usual IRF1 variants. They also showed an ability to reduce the transcriptional activity of wild-type IRF1, thereby suppressing its antitumor properties [11,12].

Wild-type IRF1 protein, in turn, contains 325 amino acids, is typically found in the cell nucleus, and can sometimes be detected in the cytoplasm [3]. It is structurally similar to other proteins of the IRF family and contains, as noted above, a DNA-binding domain (DBD) at the N-terminus and a type 2 IRF-association domain (IAD2) at the C-terminus. Between these regions, immediately after the DBD, there is the NLS region, a nuclear localization signal [13]. The IRF1 protein is also known to have two clustered phosphorylation sites—one within the region of amino acids 138–150 and another in the region of amino acids 219–231. Mutations in the sequences encoding these regions resulted in a significant decrease in IRF1 activity [14].

## 4. Role of IRF1 in IFN Signaling

IRF1 was the first to be identified among other interferon response factors. Due to its ability to bind specific DNA sequences, IRF1 was initially identified as a transcriptional activator of the IFNβ gene and several IFN-stimulated genes [15]. Currently, IRF1 is also known as a signal transduction mediator for different classes of interferons (IFNs). IFN stimulation of cells leads to JAK/STAT-mediated induction of the transcription of *IRF1* and other IFN-stimulated genes that are responsible for the primary response to IFN [16]. *IRF1* transcription is thought to be triggered by the binding of various molecular messengers to a GAS sequence (TTTCCCCGAAA) in the proximal part of its promoter [17,18].

IFN types are associated with certain downstream messengers that bind to specific nucleotide sequences (GAS or ISRE) located in regulatory regions of target genes. Among them, pSTAT1 homodimers (GAF protein complex), involved in IFN II and IFN I signaling, were shown to be able to interact with the GAS sequence and enhance *IRF1* transcription, as were pSTAT1/pSTAT2 (GAF-like complex) and pSTAT1/STAT2 heterodimers, responsible for IFN I signaling [17]. It was also demonstrated that transcription of the *IRF1* gene was dependent on IFN I and IFN II stimulation only, without reliance on class III IFNs [19]. Moreover, most cell types have been shown to activate *IRF1* transcription more successfully in response to type II IFN stimulation, compared to IFN I [20]. These data suggest that of all of the IFNs, class II IFNs (IFNγ) may be considered to be the most important regulators of IRF1 levels in cells.

The IFN-independent induction of *IRF1* gene expression was also reported. It was suggested that in cells not stimulated by IFN, *IRF1* gene expression is maintained at a low basal level by the binding of non-phosphorylated STAT1 homodimers to its GAS sequence. Then, in these non-IFN-stimulated cells, IRF1 can affect the transcription of some of its target genes, presumably to maintain a certain basal level of its products in the cell [17,21].

It was also suggested that IRF1 and STAT1 may co-regulate the transcription of some interferon-stimulated genes containing both ISRE and GAS sequences. IFNγ-treated human myeloblastic leukemia cells demonstrated a common phenomenon of IRF1 and STAT1 co-binding to ISRE and GAS-containing ISGs [22]. Variations in such cooperation of IRF1 and STAT1 were proposed as an explanation for differences in ISGs’ response between cell types [21]. In addition, *STAT1*, *STAT2*, and *IRF9* were identified as IRF1 target genes, suggesting the presence of a positive feedback loop in the IFN I and IFN II signaling pathways [17]. Further research showed that IRF1 does not activate IFNβ gene (*IFNB1*) transcription, but instead may regulate IFN type III (*IFNL1*) expression in response to RNA viruses [23].

IRF1 can be classified as one of the major mediators of IFN signaling. In this case, class II IFN—IFNγ—may be considered the main regulator of IRF1 levels in the cell. Class I IFNs are believed to be less important mediators of IRF1 expression, and the IRF1 protein itself is capable, under certain conditions, of influencing the transcription of genes encoding class III IFNs. Crosstalk in IFN signaling involving IRF1 was also described. Not only does the IRF1 level in the cell depend on pSTAT1 homodimers, but the IRF1 protein itself can affect the expression of STAT1 and several other mediator genes, thereby indicating positive feedback in IFN signaling. In general, IRF1 may be involved in the antiviral, immunomodulatory, and antiproliferative functions of IFNs, depending on the specific molecular context.

## 5. IRF1 Tumor-Suppressive Functions

The majority of the available information on IRF1’s impact on carcinogenesis mainly revolves around its antitumor properties (Figure 1, left side).

For example, the ability of IFNγ-induced IRF1 to participate in several antitumor mechanisms has been demonstrated in breast cancer cells. Cells were treated with IFNγ, after which more than 17,000 DNA binding events to IRF1 were detected, with the majority of these binding events triggering transcription of target genes. Functionally, most of these binding sites were found to be associated with *apoptosis*, *DNA damage response*, and *immune response* [24]. Another study on breast cancer cells also showed that IRF1, as part of the IFNγ and TNF-α signaling pathways, plays a role in *suppressing growth* and *inducing apoptosis* of malignant cells. IRF1 inhibited the NF-kB transcription factor (NF-kB p65) and antiapoptotic proteins FLIP, cIAP1, TRADD, TRAF2, XIAP, and survivin (BIRC5) in these cells. At the same time, increasing the amount of IRF1 in normal, non-malignant breast cell lines did not result in a significant suppression of cell growth, increased apoptosis, or changes in the amount of NF-kB p65. It was hypothesized that such differences may indicate the involvement of IRF1 as an antitumor agent in a malignant cell-specific mechanism [25]. Other evidence also suggests that IRF1 is associated with tumor cell apoptosis, in addition to the above. It can be involved in regulating apoptosis, either dependent on the p53 protein and independent of it, through various molecular interactions. IRF1 inhibits the expression of the survivin protein gene (*BIRC5*), activates the transcription of caspase genes (*CASP1*, *CASP7,* and *CASP8*), the *PUMA* apoptosis modulator gene, and others, and also has a positive effect on the induction of the TRAIL signaling pathway, known for its ability to activate tumor cell apoptosis [26,27,28,29,30,31,32].

IRF1 has also been shown to have an impact on p21-dependent *cell cycle arrest*. The effect of IRF1 on upregulating p21 expression in cells was shown in breast, lung, and gastric cancer cell lines. p21 was further shown to mediate IRF1’s ability to induce dose-dependent cell cycle arrest at the G1 phase in cancer cells, which was associated with reduced levels of CDC-2, CDK2, CDK4, cyclin B1, and cyclin E1 [33].

Another mechanism of carcinogenesis, replicative immortality, can also be negatively affected by IRF1. IRF1, as part of the IFN signaling pathway, has been shown to inhibit *hTERT* gene expression and *telomerase activity* [34].

In addition, IRF1 was shown to affect cell *proliferation*. Its ability to indirectly suppress the activity of the gene promoter of a key marker of cell proliferative activity, Ki-67, was discovered. This is thought to be due to a reduced amount of protein (but not mRNA) of the Sp1 transcription factor [35].

IRF1 is known to control the expression of a significant number of genes which are crucial for both innate and acquired immunity. In this context, IRF1 was proposed to provide a link between these two types of immunity. IRF1 has demonstrated the ability to influence the development and differentiation of NK cells, dendritic cells, CD8+ T cells, and type 1 T helper cells [31]. This phenomenon was observed in the tumor microenvironment as well. For example, IRF1 has been shown to be associated with the development of different *tumor immune phenotypes* in melanoma cell lines. In one study, out of fifteen cell lines treated with IFNγ and TNF-α, three lines with the highest and three lines with the lowest level of IRF1 activation in response to stimulation were selected. These two groups showed completely different patterns in further transcriptomic analysis. In the group with high levels of IRF1 activation, inhibition of mTOR and Wnt/β-catenin signaling pathways was observed, which was previously shown to be associated with poor prognosis and a more aggressive immune phenotype in melanoma. These results suggested an association between decreased cell IRF1 responsiveness to IFNγ and TNF-α stimulation with less favorable tumor immune phenotypes and poor prognosis [36]. The involvement of IRF1 in IFNγ-regulated MHC class I *antigen presentation* was also suggested [37]. Furthermore, both the intrinsic antitumor properties of IRF1 and its ability to influence the formation of the *tumor immune microenvironment* through the IRF1/CXCL10/CXCR3 axis were demonstrated in hepatocellular carcinoma. IRF1 increased *CXCL10* gene transcription and exhibited antiproliferative and pro-apoptotic effects on tumor cells, presumably by activating the CXCL10/CXCR3 autocrine pathway. Alternatively, IRF2, considered an IRF1 antagonist, decreased *CXCL10* gene expression. Additionally, IRF1 promoted the migration of NK and NKT cells, as well as CD4+ and CD8+ T cells, into the tumor microenvironment. Increased levels of IRF1 in tumor cells in this study were associated with increased secretion of IFNγ by infiltrating NK and NKT cells and with induction of apoptosis by the CXCL10/CXCR3 paracrine axis [38].

There is also evidence for the role of IRF1 in the inhibition of *angiogenesis*. IRF1 has been found to inhibit VEGF-stimulated proliferation, migration, and endothelial cell invasion. This antiangiogenic activity of IRF1 was found to occur without a direct cytotoxic effect on endotheliocytes and fibroblasts. IRF1 demonstrated direct binding to VEGFR2 (but not VEGFR1), inhibited the PI3K/Akt signaling pathway and eNOS phosphorylation, and prevented tumor growth and angiogenesis in vivo. The suppression of angiogenesis and downregulation of VEGF in cells in this study were found to be due to the effects of wild-type IRF1 and IRF1(Δ7), but not the other splice variants IRF1(Δ8) and IRF1(Δ9) [39].

IRF1 has also been associated with tumor *response to chemotherapy and radiotherapy*. In particular, the role of IRF1 in cisplatin-induced apoptosis was studied in lung cancer cells. Exposure of cells to cisplatin caused an increase in IRF1, mitochondrial membrane depolarization, induction of oxidative stress, and cell death by apoptosis, while inhibiting cell death by autophagy. Exclusion of the IRF1 influence, on the contrary, led to the suppression of such effects [40].

The role of IRF1 in increasing the radiosensitivity of tumor cells has been demonstrated in colorectal cancer cells. IRF1 was shown to limit cell proliferation and increase cell sensitivity to ionizing radiation. This effect was proposed to be related to its role in regulating the expression of several interferon-dependent genes. Colorectal cancer tissue samples showed reduced amounts of IRF1 compared to adjacent unaffected tissue samples, and higher IRF1 levels were associated with better prognosis [41].

## 6. IRF1 Tumor-Promoting Functions

Despite many studies indicating the antitumor effect of IRF1, it has also been shown to promote tumor growth and progression (Figure 1, right side). Some of the tumor-promoting properties of IRF1 are listed below.

The ability of IRF1 to increase programmed death ligand 1 (PD-L1) gene expression is now well established. PD-L1, a programmed cell death ligand, stimulates the PD-1 receptor on T lymphocytes, resulting in the suppression of T cell immune responses and *tumor immune escape*. Stimulating a cell with IFNγ activates the expression of the *IRF1* gene, which then causes the IRF1 protein to bind directly to the promoter of the PD-L1 gene (*CD274*) and enhance its transcription. In some cases, IRF1 can also bind to the promoter of the PD-L2 (*CD273*) ligand gene, which inhibits the antitumor immune response as well [42]. In melanoma and colon cancer cells, IRF1-deficient tumor cells exhibited reduced growth rates due to the increased cytotoxic properties of CD8 T lymphocytes. Loss of IRF1 in tumor cells suppressed PD-L1 and sensitized T cells to the immune response [43]. Later, two functionally active IRF1 binding sites (IRE1 and IRE2) were discovered in the *CD274* gene promoter, and IRF2 was shown to compete with it to bind to these sites. Stimulation of hepatocellular carcinoma cells with IFNγ led to an increase in the amount of IRF1 and, subsequently, the amount of PD-L1 in the cells. Increasing the amount of IRF2 in these cells did not alter the level of IRF1, but inhibited its ability to promote *CD274* gene expression [44]. In ovarian cancer cells, the increase in *CD274* transcription in response to IFNγ stimulation of cells was confirmed to occur under the control of the JAK1/STAT1/IRF1 signaling pathway [45]. In addition, the mechanisms associated with IFNγ-mediated upregulation of IRF1 and PD-L1, which triggers tumor evasion of T cell immunity, are also present in tumor-infiltrating myeloid-derived suppressor cells (MDSCs) [46]. These data allow us to consider IRF1 and its signaling pathways key regulators of PD-L1, which is one of the main “checkpoints” of antitumor immune responses. The measurement of cellular IRF1 levels is considered a promising biomarker for PD-L1/PD-1 inhibitor therapy, in some cases more effective than the measurement of PD-L1 levels alone [44,47,48]

There is evidence of another effect of IRF1 that involves the *suppression of antitumor immune responses*. IRF1 was found to be able to reduce the expression of the CXCL9 chemokine gene, which, among its other properties, is important for attracting T lymphocytes to the tumor site. IRF1 does not directly bind to the *CXCL9* gene promoter, but regulates its expression indirectly. It was found that IRF1 binds to the promoter of the *SOCS1* (suppressor of cytokine signaling 1) gene and enhances its expression, and the SOCS1 protein, in turn, is able to suppress the JAK/STAT signaling pathway. Thus, increasing the amount of SOCS1 protein in cells reduces STAT1 phosphorylation, inhibits STAT1-dependent signaling pathways, and reduces the expression of many target genes, including the chemokine gene *CXCL9*. In this manner, IRF1 and SOCS1 form a negative feedback loop in IFNγ signaling. The suppression of IRF1 and SOCS1, in contrast, resulted in increased expression of *CXCL9* and activation of the T cell immune response. As noted, the intensity of this effect may vary depending on the cell type and exposure to other external stimuli [49].

Some evidence suggests that IRF1 may be involved in the development of *resistance to certain chemotherapeutic drugs* in tumors. In contrast to the antitumor role of IRF1 mentioned a few paragraphs above, which is associated with its involvement in the cellular response to cisplatin, there is evidence of another connection between IRF1 and cisplatin. Cisplatin exposure was found to increase *IRF1* expression in ovarian cancer cells, resulting in p21-dependent cell cycle arrest and a further paradoxical decrease in sensitivity to cisplatin. This effect may be explained by a decrease in cell proliferative potential [50]. IRF1 activity in ovarian cancer cells was also associated with resistance to paclitaxel, another chemotherapeutic agent [51].

The role of IRF1 in inducing *epithelial–mesenchymal transition* (EMT) in tumor cells has also been reported. These findings were associated with increased growth and metastatic potential of basal-like breast cancer cells [52]. However, information on the connection between IRF1 and the epithelial–mesenchymal transition of tumor cells remains incomplete. EMT was also shown to be associated with IRF1 downregulation [53].

As demonstrated, there is a dichotomy in the way IRF1 affects tumors. On the one hand, its antitumor properties have been clearly demonstrated, such as inducing apoptosis, cell cycle arrest, modulating the tumor immune microenvironment, suppressing cell proliferative activity, suppressing angiogenesis, suppressing telomerase activity, and increasing the sensitivity of cells to radiation therapy. On the other hand, its opposite property that can lead to increased tumor growth is also evident—IRF1 activity in cells can lead to inhibition of the antitumor immune response through PD-L1 upregulation. There is also evidence that the antiproliferative and pro-apoptotic properties of IRF1 may be involved in the development of tumor resistance to several antitumor chemotherapeutic drugs, which are known to act more effectively on actively dividing cells. These apparent dual and contradictory roles of IRF1 are sometimes considered evidence of a central and integrative role between certain essential mechanisms of carcinogenesis and antitumor immune responses.

A similar dual effect on tumor progression has been extensively described for the IFNγ signaling pathway in general. Various pro- and antitumor effects are continuously being studied and systematized [54,55,56]. IFNγ regulates many antitumor mechanisms in cells and can affect processes such as apoptosis of tumor cells, changing their antigenic properties, remodeling the tumor microenvironment, and improving the response to immunotherapy, and many other processes. Its opposite effects are equally well known. IFNγ also suppresses the antitumor immune response, increases the metastatic potential of tumor cells, enhances angiogenesis, reduces the migration of cytotoxic T lymphocytes, and more. IFNγ was also shown to influence all stages of tumor “immunoediting” and control the balance between tumor promoters and suppressors in the tumor microenvironment, which is why it is considered a potential biomarker for cancer immunotherapy [57]. The possible reasons for this dual effect of IFNγ signaling on tumor growth and development are still debated. To determine the specific role of IFN and its signaling pathways in tumor progression, it is proposed to investigate its effects in a wide range of cellular, temporal, and molecular settings [54,57]. Furthermore, it has been suggested to explore alterations in different parts of the IFNγ signaling pathway—both in the IFNGR/JAK/STAT axis and in the expression levels of various downstream target genes [58].

As mentioned above, IRF1 activity in the cell is predominantly under the control of class II IFN, IFNγ. Although the activation of IRF1 is not responsible for all of the effects of IFNγ, its activity influences many important downstream targets. The clearly established association between IRF1 and PD-L1, as well as its ability to influence many essential mechanisms of carcinogenesis, certainly make it a promising target for further research. Identification of the factors determining the tumor-specific activity of IRF1 in a specific context may help not only improve our understanding of the mechanisms of the antitumor immune response, but also identify specific prospects for the development of new biomarkers or targets for clinical practice.

## 7. Mechanisms of IRF1 Regulation in Cancer

The literature has accumulated a large amount of data on the regulation of IRF1 at different molecular levels. Although the mechanisms that change the activity or amount of IRF1 in cells are complex and diverse, more detailed studies may bring us closer to defining the main pathway that regulates IRF1 activity in tumor cells and the tumor microenvironment. The mechanisms of IRF1 regulation associated with the formation of various neoplasms are outlined below (Figure 2).

### 7.1. IRF1 Gene Alterations

The regulation of IRF1 at the genetic level is not well understood. Nonetheless, there is still some amount of research in the literature that links the occurrence of abnormalities in the primary structure of *IRF1* DNA with the development of certain malignant tumors. For example, deletion of the 5q31.1 locus within the *IRF1* gene and the occurrence of its inactivating mutations were associated with the development of acute myeloid leukemia and myelodysplastic syndrome in patients [59]. Additionally, loss of heterozygosity at the same 5q31.1 locus was observed in esophageal cancer and *BRCA1*-mutant breast cancer samples [60,61]. Some *IRF1* point mutations are also known to occur in malignant tumors. In non-small cell lung cancer cells, the W11R amino acid substitution was found in a variant of the IRF1 protein that failed to bind to DNA [62]. IRF1 variants with K75E and E222K substitutions found in chronic myeloid leukemia also had reduced transcriptional activity and DNA-binding ability [63]. In gastric adenocarcinoma, M8L substitutions were found to reduce the transcriptional activity of IRF1 but did not affect its DNA binding activity [64]. Finally, a single nucleotide polymorphism A4396G of the *IRF1* gene was described in breast cancer cells [65]. However, the importance of these mutations for the development of malignancies and the specific role of altered IRF1 variants in the carcinogenesis are not fully understood.

### 7.2. Transcriptional and Epigenetic Regulation of IRF1

The importance of methylation of the *IRF1* gene promoter as a mechanism of its regulation in malignant cells currently cannot be clearly determined. However, there are several things that should be taken into account. In a study of sixteen different gastric cancer cell lines, *IRF1* methylation was not detected in any of them, although its expression levels were generally lower in them compared to normal samples [66]. *IRF1* methylation levels in non-small cell lung cancer (NSCLC) tissues were not statistically different from those in adjacent tissue samples. However, in cases where *IRF1* methylation levels were still elevated, PD-L1 levels were reported to be significantly decreased. IRF1 and PD-L1 levels in cell lines increased significantly after further use of methylation inhibitors [67]. It appears that the methylation level of the *IRF1* promoter can still be modified by some additional factors in different cellular contexts. For example, overexpression of the methyltransferase gene DNMT3B led to increased *IRF1* gene methylation levels and repression of its expression [68]. In contrast, exposure of cells to the TET1 dioxygenase under IFN-γ-stimulated conditions could demethylate the *IRF1* gene and activate its transcription. This mechanism, including the NAD+-dependent increase in TET1 activity and further activation of the IRF1-PD-L1 axis, was associated with decreased antitumor immune responses in hepatocellular carcinoma tissue samples [69].

Some epigenetic effects may also contribute to changes in *IRF1* expression. For example, EZH2 methyltransferase activity in hepatocellular carcinoma cells caused H3K27me3 histone modification near *IRF1* and *CD274* promoter regions, which suppressed their expression without affecting the IFNγ/STAT1 signaling pathway [70]. Another study demonstrated the ability of the MUC1-C protein to stimulate *IRF1* expression. This feature was associated both with an increase in the IFNGR1 receptor gene expression and the activity of the downstream STAT1/IRF1 pathway, and with an increase in chromatin accessibility in the *STAT1* and *IRF1* gene regions [71].

Additionally, several transcription factors are known to influence *IRF1* gene expression in cancer. The ability of the transcription factor FOXP1 to directly bind to the *IRF1* gene promoter and increase its transcriptional activity was demonstrated in pancreatic cancer [72]. A similar effect was observed for the transcription factor FOXM1c in esophageal cancer cells. FOXM1c bound to the *IRF1* promoter and increased its transcription, and higher levels of FOXM1c and IRF1 were associated with a poor survival rate in this study [73]. The NF-κB factor was also able to directly increase *IRF1* transcription in response to DNA double-strand break formation in multiple myeloma cells [74]. Another transcription factor, PAX2, was shown to increase *IRF1* transcription and induce apoptosis in ER-positive breast cancer cells in response to tamoxifen treatment [75].

In contrast, some proteins can inhibit *IRF1* expression. For example, it was found in prostate cancer cells that ARTD9 and DTX3L proteins can act as suppressors of *IRF1* gene transcription by interacting with the STAT1β transcription factor in its promoter region. This effect was accompanied by increased cell proliferation and the development of cell resistance to chemotherapy and radiotherapy [76]. The SAMD1 protein, which was upregulated in hepatocellular carcinoma cells, can also inhibit *IRF1* transcription [77]. There is also evidence for the ability of the PI3K-AKT signaling pathway to inhibit *IRF1* expression in the cell, although the precise molecular mechanisms underlying this effect are still under investigation [78,79].

### 7.3. Post-Transcriptional Regulation of IRF1

In addition to the above, several potential mechanisms of IRF1 regulation at the post-transcriptional level have been identified. Several pathologically significant alternative splicing isoforms of IRF1 pre-mRNA were discovered. Thus, the splicing factor SFPQ mediates the formation of an alternative IRF1 isoform, the IRF1Δ7 protein lacking exon 7, in T helper type 1 (Th1) cells. This alternative isoform negatively affects the antitumor properties of Th1 by inhibiting the wild-type IRF1 transcriptional activity [12]. IRF1 variants lacking exons 2 and 3 were detected in patients with acute myeloid leukemia and myelodysplastic syndrome, while the number of full-length IRF1 variants were reduced [10]. IRF1 isoforms without exons 7, 8, 9, and their combinations, as described above, were found in cervical cancer [11].

Many microRNAs were found to inhibit IRF1 mRNA levels in various tumor cells. These include miR-383, mir-130b, miR-21, miR-301a, miR-31, and others. An increase in the amount of all of the above microRNAs, except mir-130b, was associated with suppression of IRF1 antitumor effects [80,81,82,83,84]. Antisense long non-coding RNA IRF1-AS was also mentioned as a potential mediator of cancer pathogenesis. However, information regarding its specific role in cancer remains controversial [85,86].

The ability of IRF1 mRNA to undergo methylation of adenosine residues has been demonstrated. RNA-methyltransferases METTL3/14 modulated this specific modification of IRF1 and STAT1 mRNAs, which was associated with an altered response of colorectal cancer cells to PD-1 inhibitor therapy [87].

### 7.4. Post-Translational Modifications of IRF1

#### 7.4.1. IRF1 Phosphorylation

The IRF1 protein in the cell is capable of undergoing several post-translational modifications. Some of these modifications are associated with cancer progression. For example, IRF1-inactivating phosphorylation has been demonstrated in breast cancer cells. In these cells, the ability of nuclear factor IKK-ε (inhibitor of nuclear factor kappa B kinase epsilon) to phosphorylate the IRF1 protein at the C-terminus was demonstrated. Such forms of IRF1, phosphorylated at Ser215, Ser219, and Ser221, were characterized by low activity, instability, and a high frequency of degradation by the ubiquitin–proteasome pathway. Inhibition of IKK-ε restored IRF1 activity and suppressed cell proliferation and migration [88].

Activating phosphorylation of IRF1 associated with other regions of the protein has also been demonstrated. Phosphorylation of tyrosine residues was thought to be necessary for IRF1 activation and more efficient DNA binding [7]. Later, other IRF1 phosphorylation sites were discovered. For example, the protein kinase GSK3 (glycogen synthase kinase 3) was shown to phosphorylate IRF1 simultaneously at the Thr181 and Ser185 sites. This phosphorylation, occurring on the DNA-binding protein form, was indicated to be important for IRF1 transcriptional activity. Such forms of IRF1, phosphorylated at Thr181 and Ser185, although associated with the greater transcriptional activity of the protein, are, however, less stable and often become ubiquitinated by the Fbxw7α protein. It is suggested that this mechanism may be important for the timely removal of IRF1 from the target gene DNA and the acceleration of subsequent transcription cycles [89].

#### 7.4.2. IRF1 Ubiquitination

Ubiquitination is also believed to be an important mechanism for regulating the amount and activity of the IRF1 protein in cells. IRF1 is known for its ability to undergo K48-linked polyubiquitination and subsequent proteasomal degradation, which is thought to be dependent on a region of the protein containing 39 C-terminal amino acid residues [90]. Later, other mechanisms of IRF1 ubiquitination were discovered. The CHIP ubiquitin ligase was demonstrated to induce IRF1 mono- and polyubiquitination in the DNA-binding domain, which occurred by both K48-linked and K63-linked mechanisms [91]. Mono-ubiquitination of IRF1 at residues Lys39, Lys50, or Lys78 of its DNA-binding domain can be induced by the MDM2 ligase. Unlike polyubiquitination, this modification did not activate proteasomal degradation of the IRF1 protein. This study also showed that DNA-bound forms of IRF1 are unable to undergo ubiquitination by CHIP and MDM2 ligases, unlike its free forms [92]. SPOP (speckle-type POZ protein) can also cause polyubiquitination and further degradation of the IRF1 protein [93]. However, polyubiquitination of IRF1 is not always associated with its degradation. Polyubiquitination of the IRF1 DBD by a K63-linked mechanism involving cIAP2 and S1P proteins did not cause degradation of IRF1, but rather activated and increased the transcription of target genes [94]. Apparently, the effect of IRF1 ubiquitination may vary depending not only on the number of ubiquitin monomers attached, but also on the ubiquitination mechanism, the presence of DNA-bound forms of IRF1, and the involvement of certain sites in the protein itself.

#### 7.4.3. IRF1 SUMOylation

Sumoylation of the IRF1 protein, known as the process of binding SUMO (small ubiquitin-like modifier) protein molecules to its lysine residues, occurs at the same sites at the C-terminus of IRF1 as ubiquitination. Sumoylation is thought to protect the IRF1 protein from polyubiquitination and subsequent protein degradation at these sites [95]. This modification may occur, as it was found, under the influence of Ubc9 (ubiquitin carrier protein 9), a protein that can directly interact with proteins of the IRF family [96]. Sumoylated forms of the IRF1 protein were shown to be more stable, but had decreased transcriptional activity [95,96]. The co-expression of IRF1 and SUMO1 resulted in the repression of IRF1 transcriptional activity, which negatively affected its ability to induce cell apoptosis [95].

#### 7.4.4. IRF1 Acetylation and Methylation

Acetylation of the IRF1 protein in malignant cells is also known. The ability of the acetyltransferase KAT8 to directly interact with IRF1 and specifically acetylate it at the K78 lysine residue has been demonstrated. This acetylated form of IRF1 increased its DNA-binding activity, resulting in the upregulation of PD-L1. A positive feedback loop was also shown for this mechanism. Acetylation of IRF1 led to additional involvement of KAT8 and acetylation of histone H4K16 in the PD-L1 (CD274) promoter region. In contrast, disruption of this connection between IRF1 and KAT8 resulted in decreased cellular PD-L1 levels, decreased tumor mass, and increased numbers of tumor-infiltrating CD8+ T cells [97]. The ability to deacetylate IRF1 was demonstrated by SIRT1, a histone deacetylase whose activity change is associated, among other things, with tumor progression [98,99].

The ability of the IRF1 protein to undergo methylation at several lysine sites under the action of protein methyltransferase SET7/9 (KMT7) was also revealed. The functional characteristics of such methylated forms of IRF1, as well as their role in cancer progression, remain unknown [100].

#### 7.4.5. Other Interactions

The intracellular activity of IRF1 can also be modified through its interaction with several other proteins. For example, IRF1 can be co-activated by the Smad7 protein, which increases its binding to target genes such as the CASP8 gene, thus increasing the pro-apoptotic activity of IRF1 [101]. Moreover, it was demonstrated that another IRF, IRF8, can enhance IRF1’s ability to induce apoptosis [102]. JNK1 kinase can also upregulate IRF1 in the cell, although not by directly interacting with the protein itself [103]. In contrast, IRF1 activity in the cell may be reduced as a result of interaction with certain other proteins. ZBED2, a DNA-binding protein, both directly inhibited IRF1 transcriptional activity and competed with it for binding sites on target genes [104]. The Wnt/β-catenin signaling pathway has also been identified as a potential negative regulator of IRF1. In a previous study, the Wnt/β-catenin signaling pathway decreased the activity of the USP1/UAF1 deubiquitinase complex, which resulted in increased IRF1 degradation via the ubiquitin–proteasome pathway [105]. Another possible mechanism could be the effect of the Wnt/β-catenin pathway on the change in IRF1 intracellular localization under IFNγ and TNFα stimulation [36].

Lastly, variations in IRF1 distribution between the cytoplasm and nucleus of cells may lead to variations in IRF1 activity. An example of this effect was demonstrated by the STXBP6 protein, which is involved in the SNARE complex’s assembly. STXBP6 was shown to bind to IRF1 and prevent its translocation into the nucleus, hence preventing the transcriptional activation of target genes. STXBP6-dependent cytoplasmic retention of IRF1 then resulted in reduced PD-L1 levels, improved T cell antitumor immune response, and reduced tumor mass. Similarly, lowering cellular STXBP6 levels promoted nuclear translocation of IRF1 and improved cellular sensitivity to IFNγ activation [106].

All of the abovementioned IRF1’s tumor-suppressive and tumor-promoting effects reported in different cancer contexts, as well as the suggested mechanisms underlying them, are listed in Table 1.

## 8. Conclusions

Without a doubt, IRF1 can be considered one of the most significant mediators of IFNγ action, both in tumor cells and in the cells of their microenvironment. It appears that the intrinsic effects of the IRF1 protein on carcinogenesis and tumor growth are as ambivalent as the known functions of the IFNγ signaling pathway itself. On the one hand, IRF1 can influence a number of pathways that result in the suppression of tumor growth. For instance, it is well known that it has an impact on apoptosis induction, cell cycle arrest, formation of the tumor immune microenvironment, and many other antitumor mechanisms. On the other hand, IRF1 is also known to negatively affect antitumor immunity, mainly through the upregulation of “immune checkpoint” molecules, in particular PD-L1 and PD-L2. Moreover, direct contradictions between the pro- and antitumor activities of IRF1 are frequently seen in the literature. It is yet unknown what precise factors could definitively account for these specific differences in IRF1’s impact on malignant tumors. Elevated levels of IRF1 in cells under different conditions may be associated with both tumor progression and inhibition of tumor growth. Various theories have been proposed to explain this phenomenon, including the impact of distinct tissue and cellular contexts, different times of exposure to certain processes, and a range of molecular interactions involving IRF1 within cells. It is also suggested that this uncertainty should be considered a possible opportunity to explore the mechanisms that maintain the balance between tumor-suppressive and oncogenic processes in tumor cells and their microenvironment. Previously, a similar duality has been studied for the main upstream regulator of IRF1, IFNγ. Research on interactions involving IRF1 could explain more profound mechanisms for such confusing effects of IFNγ on tumor growth.

Studying IRF1 in particular and IFNγ signaling pathways in general may not only help to clarify several fundamental mechanisms but may also allow for the use of IRF1 and related pathways as predictors of response to PD-1/PD-L1 inhibitor therapy. As previously indicated, numerous studies have clearly demonstrated that the IFNγ-IRF1-PD-L1 axis is involved in controlling the response to immunotherapy in many tumors. Currently, there is an ongoing search for the most reliable biomarkers to predict tumor response to immune checkpoint inhibitor therapy, the risk of immune-related adverse events during treatment, and the development of secondary resistance to these drugs [107,108]. In this context, identifying key mechanisms for IRF1 regulation during tumorigenesis may help determine the prospects for using IRF1 (alone or in combination with other molecules) as a biomarker or target for cancer therapy.

## Figures and Tables

**Figure 1 ijms-25-02153-f001:**
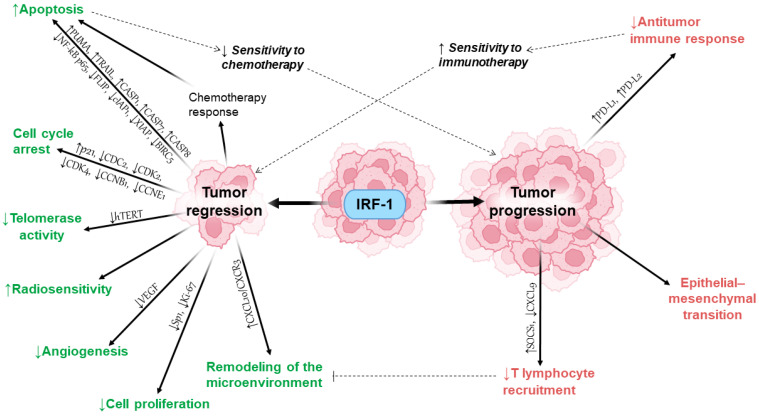
Tumor-suppressive and tumor-promoting functions of IRF1. An altered response to certain chemotherapeutic drugs, such as paclitaxel and cisplatin, has been associated with some of IRF1’s anticancer effects. Alternatively, its ability to upregulate immune checkpoint molecules (mainly PD-L1) and thus promote tumor immune escape is thought to modulate tumor responses to immune checkpoint blockade therapy.

**Figure 2 ijms-25-02153-f002:**
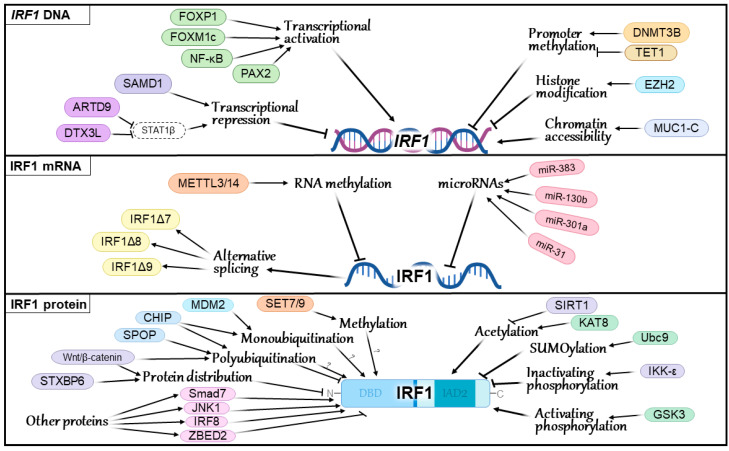
IRF1 regulation mechanisms in cancer. Question marks indicate modifications with as yet unknown effects on the precise functional characteristics of the IRF1 protein.

**Table 1 ijms-25-02153-t001:** IRF1 effects reported in different cancer types.

IRF1 Effect	Cancer Type	Proposed Mechanisms	References
**Tumor-suppressive**	Breast	Apoptosis, immune response, and DNA damage response pathways; Suppression of antiapoptotic proteins (NF-kB, FLIP, cIAP1, TRADD, TRAF2, XIAP, and BIRC5); Pro-apoptotic protein upregulation (PUMA, TRAIL, and caspase-1 and -8); caspase-3, -7, and -8 cleavage; Cell cycle arrest (CDK-2, -4, CDC2, CCNB1, CCNE1 downregulation, and p21 upregulation); Apoptosis induction in response to tamoxifen treatment; Reduced cell proliferation and migration potential.	[24,25,26,27,29,30,32,33,75,88,101]
	Cervical	Telomerase (hTERT) downregulation.	[34]
	Cholangiocarcinoma	Suppression of tumor cell proliferation, migration, and invasion.	[80]
	Colorectal	Inhibition of cell proliferation, G1 cell cycle arrest, induction of apoptosis, and increase in cell radiosensitivity. Induction of apoptosis (caspase-3 activation) and reduction in cell growth.	[41,105]
	Esophagus (ESCC)	Inhibition of tumor cell proliferation and promotion of apoptosis (mediated by an IRF1 antisense RNA).	[85]
	Ewing’s sarcoma	Apoptosis (caspase-7 activation and Bcl-2 downregulation).	[28]
	Gastric	Apoptosis (PUMA upregulation); Cell cycle arrest (p21 upregulation).	[30,33]
	Hepatocellular	Formation of antitumor immune microenvironment (CXCL10/CXCR3 autocrine pathway).	[38]
	Lung (NSCLC)	Cell cycle arrest (p21 upregulation); MHC class I antigen presentation; Cisplatin-induced apoptosis (caspase 3 activity); Macrophage polarization.	[33,37,40,82]
	Multiple myeloma	Increase in APCs’ presentation activity in response to chemotherapy (CIITA upregulation).	[74]
	Neuroblastoma	Increase in T cell recruitment (upregulation of chemokines CXCL9 and CXCL10).	[79]
	Pancreatic	Inhibition of tumor cell growth and invasion; Tumor cell growth arrest.	[72,104]
	Prostate	Inhibition of tumor cell proliferation and induction of apoptosis.	[76]
	Renal cell	Anti-proliferative effect (Sp1-mediated Ki-67 downregulation).	[35]
	Skin melanoma	Tumor immune phenotype regulation (mTOR and Wnt/β-catenin pathways).	[36]
**Tumor-promoting**	Breast	Suppression of an antitumor immune response (SOCS1-mediated CXCL9 downregulation); Epithelial–mesenchymal transition.	[49,52]
	Colorectal	Tumor immune escape (PD-L1 upregulation).	[43,106]
	Endometrial	Tumor immune escape (PD-L1 upregulation).	[93]
	Esophagus (ESCC)	Promotion of tumor cell invasion and migration.	[73]
	Hepatocellular	Tumor immune escape (PD-L1 upregulation); Tumor cell migration and invasion.	[44,69,70,81]
	Ovarian	Tumor immune escape (PD-L1 upregulation); Suppression of an antitumor immune response (SOCS1-mediated CXCL9 downregulation); Resistance to cisplatin and paclitaxel.	[45,49,50,51]
	Prostate	Inhibition of T cell functions (IDO1, WARS, PTGES, ISG15, and SERPINB9 upregulation).	[71]
	Skin melanoma	Tumor immune escape (PD-L1 and PD-L2 upregulation).	[42]

## Data Availability

Not applicable.

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
