# Peer review of "Roles of Interferon Regulatory Factor 1 in Tumor Progression and Regression: Two Sides of a Coin"

_ijms, 2024, doi:10.3390/ijms25042153_

Round 1

Reviewer 1 Report

Comments and Suggestions for Authors

In this manuscript, the authors comprehensively overview the role of Interferon Regulatory Factor 1 (IRF1) in various tumorigenic contexts, including both sides of tumor progression and tumor regression. Also, the authors kindly introduce all the information on IRF1 from the bottom to details in various cancer contexts and its regulation mechanisms. This review includes much beneficial information for the reader or researcher interested in IRF1. I would like to express sincere respect for the authors’ efforts to summarize the many papers on IRF1 studies. 

This manuscript is well-organized and has comprehensive information. I just want to give one comment to improve this manuscript to be more attractive for the IJMS readers.

  1. Overall, the authors overview the role of IRF1 both in tumor progression and tumor regression. However, much information was included, making it difficult to understand the direction of action in different cancer contexts. For example, IRF1 is tumor progression or suppression in Colon cancer? Additionally, what are the suggested mechanisms (Reference)? 

Therefore, it would be nice to add a summarized chart or figure for the tumor progression or suppression roles of IRF1 in each cancer type. This information may be what readers want to know from the reviews regarding IRF1.

Comments on the Quality of English Language

Minor editing of English language required

Author Response

Response to Reviewer 1

Dear reviewer,

Thank you very much for evaluating our manuscript. We appreciate the time and effort that you dedicated to providing feedback on our manuscript, and are sincerely grateful for your encouraging and valuable comments. They are of great assistance to us for improving and revising our manuscript. We have taken your thoughtful suggestions fully into account in revision. We are submitting the corrected manuscript with all modifications highlighted in red.

The manuscript has been revised, and our responses to the comments are as follows:

  1. Overall, the authors overview the role of IRF1 both in tumor progression and tumor regression. However, much information was included, making it difficult to understand the direction of action in different cancer contexts. For example, IRF1 is tumor progression or suppression in Colon cancer? Additionally, what are the suggested mechanisms (Reference)?

Therefore, it would be nice to add a summarized chart or figure for the tumor progression or suppression roles of IRF1 in each cancer type. This information may be what readers want to know from the reviews regarding IRF1.

Response - Thank you so much for your precious suggestion. A summarized chart would certainly help to improve our manuscript and provide readers or researchers interested in IRF1 with more structured information about IRF1 actions in different cancer contexts. We have now added Table 1 to Section 7.4.5, which includes IRF1 effects from all cancer types discussed.

Some minor edits have also been made in the English language.

Thank you again for your kind words and your time in reviewing our manuscript.

Yours sincerely,

Reviewer 2 Report

Comments and Suggestions for Authors

This thoughtful review article by Perevalova and colleagues aggregates the current understanding of the transcription factor IRF1's complex involvement in cancer pathology. Masterfully highlighting IRF1's dichotomous tumor-suppressive and tumor-promoting effects, the authors underscore that IRF1 seems to finely balance pro- and anti-tumor mechanisms. Through coverage of regulatory influences on multifaceted IRF1 activity, they note that further elucidating precise control mechanisms could elucidate IRF1’s conflicting cancer roles. This paper represents a very interesting topic and should have a broad audience. I just have several comments which may improve the manuscript:

1.  Does the introduction provide sufficient background and include all relevant references?

2. The text in the figures appears to be quite small, and certain words are blurry, making it difficult for readers to comfortably read. It is suggested to increase the font size to enhance readability for readers.

Author Response

Dear reviewer,

Thank you very much for your time and effort in reviewing our manuscript and providing valuable and constructive comments. We sincerely appreciate your encouraging words and insightful comments. They have been very useful in improving the current version of the manuscript. We have studied your comments carefully and have made corrections in line with the suggestions. We are submitting the corrected manuscript with all modifications highlighted in red.

The manuscript has been revised, and our responses to the comments are as follows:

  1. Does the introduction provide sufficient background and include all relevant references?

Response - Thank you very much for your comment. The introduction has been supplemented with some important background information about IRF1, with the references included for readers interested in IRF1 studies.

  1. The text in the figures appears to be quite small, and certain words are blurry, making it difficult for readers to comfortably read. It is suggested to increase the font size to enhance readability for readers.

Response - Thank you so much for pointing this out. We have edited figures 1 and 2, and increased the font size to make it more comfortable to read.

Thank you again for your kind words and your time in reviewing our manuscript.

Yours sincerely,